# Extensive breaking of genetic code degeneracy with non-canonical amino acids

Clinton A. L. McFeely[1,2], Bipasana Shakya [1,2], Chelsea A. Makovsky[1,2], Aidan K. Haney[1], T. Ashton Cropp[1] & Matthew C. T. Hartman [1,2] ✉

Genetic code expansion (GCE) offers many exciting opportunities for the creation of synthetic organisms and for drug discovery methods that utilize in vitro translation. One type of GCE, sense codon reassignment (SCR), focuses on breaking the degeneracy of the 61 sense codons which encode for only 20 amino acids. SCR has great potential for genetic code expansion, but extensive SCR is limited by the post-transcriptional modifications on tRNAs and wobble reading of these tRNAs by the ribosome. To better understand codon-tRNA pairing, here we develop an assay to evaluate the ability of aminoacyl-tRNAs to compete with each other for a given codon. We then show that hyperaccurate ribosome mutants demonstrate reduced wobble reading, and when paired with unmodified tRNAs lead to extensive and predictable SCR. Together, we encode seven distinct amino acids across nine codons spanning just two codon boxes, thereby demonstrating that the genetic code hosts far more re-assignable space than previously expected, opening the door to extensive genetic code engineering.

Improvements in genetic code expansion (GCE) have ushered in a new era of protein biotechnologies, permitting the translation of designer proteins with site-specific modifications and the creation of highly diverse peptide libraries with non-canonical amino acids (ncAAs)[1–5]. Numerous GCE methods for the introduction of ncAAs exist, including stop codon suppression, translation of quadruplet codons and unnatural base pairs, and sense codon reassignment (SCR)[6–12]. While stop codon suppression is the most commonly used GCE method, it is the most limited due to the existence of just three stop codons. Use of quadruplet codons and unnatural base pairs each holds exciting promise, but the poor efficiency and technical hurdles for adoption of these technologies have limited their applications. Sense codons comprise 61 of the codons in the genetic code, yet they encode just 20 amino acids due to synonymous coding in groups of two, three, four, and six-fold degenerate codon boxes[13]. Complete elimination of codon degeneracy could enable the introduction of dozens of ncAAs, radically expanding the chemical repertoire of the genetic code. Whereas most in vivo examples of SCR describe the addition of a single ncAA, the flexibility of in vitro systems frequently permits the addition of 3–4

ncAA through SCR[14–17]. The customizable PURE in vitro translation system in particular has been used to introduce a code that contains 25 monomers via SCR[16]. While these advancements help illustrate the immediate potential of SCR as a GCE tool, the possibility of a non-degenerate genetic code with a massive amino acid repertoire remains out of reach due to limitations imposed by the ribosome and its recognition of tRNAs.

In particular, maximal SCR will require fully breaking the degeneracy of codon boxes[15,18,19]. This poses a significant challenge to the ribosome, as the discrimination between tRNAs is the most error-prone process in protein translation[20,21]. Furthermore, many tRNAs display expanded codon recognition, pairing successfully with codons they are not perfectly complementary towards due to post-transcriptional modifications (PTM) within the anticodon stem-loop[22–25]. Notable amongst these PTMs in *E. coli* is the 5-oxyacetic acid modification of uridines ($cmo^5U_{34}$) at the wobble position of certain tRNA isoacceptors[26–28]. The $cmo^5U_{34}$ is prevalent in the highly degenerate leucine, valine, proline, serine, threonine, and alanine codon boxes. These challenges are likely amplified upon incorporation

[1]Department of Chemistry, Virginia Commonwealth University, 1001 W Main St., Richmond, VA 23284, USA. [2]Massey Cancer Center, Virginia Commonwealth University, 401 College St., Richmond, VA 23219, USA. ✉e-mail: mchartman@vcu.edu

of unnatural amino acids, which can hinder binding affinity to EF-Tu leading to diminished translational accuracy and efficiency[29,30]. Taken together, both PTMs and challenges in ribosomal discrimination of tRNAs bearing ncAAs resist aggressive attempts at SCR.

While the overlapping codon reading rules for tRNAs have been well established, that does not mean that each tRNA isoacceptor will read a given codon with equal efficiency[22,31]. For maximal SCR, it would be helpful to have a ranking of the ability of each tRNA to read each codon. To address this open question, we focus on the most complicated codon box in *E. coli*, the 6-fold degenerate leucine codon box (comprised of a four-fold degenerate unsplit box and a two-fold degenerate split box), which is read by five different tRNAs with multiple overlapping codon reading preferences. We design in vitro translation assays in which the five leucyl tRNA isoacceptors were aminoacylated with isotopically labeled leucines containing unique masses and placed in competition with each other for each of the six leucine codons. Our assay is performed with fully modified tRNA isolated from *E. coli* total tRNA (hereafter referred to as wild-type tRNA, (wt tRNA)), in vitro transcribed tRNA lacking PTMs (t7tRNA), or mixtures of the two. Our results uncover some surprising and unreported codon reading in particular by wt tRNAs. Moreover, when we use a hyperaccurate ribosome mutant, we are able to reassign the six-fold degenerate leucine codon box to encode five distinct amino acids in a single experiment. Finally, we apply our orthogonality rules to the four-fold degenerate valine codon box, demonstrating predictable SCR. In total, we demonstrate the reassignment of the 10 leucine and valine codons to encode for seven distinct amino acids, thereby providing evidence that the standard genetic code has more re-assignable space than previously imagined.

## Results

### Development of a competitive codon reading assay to define the boundaries of SCR

The rules regarding which tRNAs read which codons were proposed long ago, with the permissibility of so-called "wobble pairs"[32]. These readthrough patterns have since been quantitatively validated using individual tRNAs;[22] however, for optimal SCR, a more actionable metric would be a ranking of the ability of tRNAs to read a given codon in competition with each other. We envisioned that such a system could be achieved if one could uniquely distinguish and quantify the translation products of each competitor tRNA by mass spectrometry. Focusing on the six-fold degenerate leucine codon family, we designed an assay where individual tRNA Leu isoacceptors were charged with a leucine isotopologue of unique mass. Each AA-tRNA was added to a custom reconstituted "protein synthesis using recombinant elements" (PURE) translation reaction[14] at equal concentration and was allowed to compete with the other tRNAs in translation reactions containing one of six mRNAs, each containing a single leucine codon (Fig. 1a). We prepared and tested two leucyl tRNA mixtures: wt tRNAs, which were purified to homogeneity from a mixture of *E. coli* total tRNA using fluorous capture[33], and t7tRNAs, which were prepared using in vitro transcription and therefore lacked all PTMs[33].

To make certain that the tRNA mixtures were added to translation at equal concentrations, we analyzed the combined AA-tRNA mixture using a MALDI-MS assay for tRNA charging[34]. The five AA-tRNAs added in each of the assays only slowed slight deviations from the expected 1:1:1:1:1 mixture (Fig. 1b). With this established, we tested the tRNA mixtures with all 6 leucine codons, analyzing the peaks by MALDI-MS. The results are shown numerically as a heatmap in Fig. 1c, and representative MALDI-MS spectra are shown in Fig. 1d and Supplementary Fig. 1. The expected codon reading patterns of the tRNAs based on the literature are highlighted with boxes: black for pairs with perfect Watson–Crick complementarity and gold for reported pairings that are not fully complementary[22,35]. Results using the modified (wt tRNA) mixture mostly followed the expected readthrough patterns: the tRNA

isoacceptor with the highest readthrough of each codon was the one forming Watson–Crick base pairs with the codon (Fig. 1c). Of the three (CUU, CUG, UUG) leucine codons reported to be "shared" between two tRNA isoacceptors[17], the CUG and CUU codons showed the expected sharing patterns, however Leu)AA with its 5-carboxymethylaminomethyl-2′-O-methyl uridine modified codon proved to be a poor competitor for the UUG codon (Fig. 1c). The A-ending codons both showed unexpected readthrough: tRNAs LeuCAG and LeuGAG in the case of the CUA codon, and tRNA LeuBAA (B = 2′-O-methyl cytidine) in the case of the UUA codon (Fig. 1c). In general, the wt tRNA mixture revealed significant codon overlap; an unattractive outcome for SCR applications for the leucine codon box. In contrast, the t7tRNA mixture that lacks PTMs revealed decreased readthrough by the wobble pairing tRNAs for the codons shared by wt tRNAs (CUG, CUU, UUG) (Fig. 1c). Universally across all six codons, the readthrough of tRNA LeuUAG decreased, highlighting the importance of its PTMs, which include the 5-oxyacetic acid uridine (cmo5U) modification[28]. As was observed with the wt tRNA mixture, the CUA codon was ambiguously read by non-cognate tRNAs LeuCAG and LeuGAG, to the point where the Watson–Crick pairing tRNA LeuUAG no longer was the overall winner of this codon (Fig. 1c). The CUG codon also had a shared reading pattern, but to a lesser extent. In summary, the t7tRNA mixture showed a reduction in codon sharing and non-specific readthrough, with the exception of the CUA codon, which was ambiguously read using both tRNA mixtures.

### Hyperaccurate ribosomes lead to improved codon orthogonality by minimizing near-cognate interactions

The high degree of similarity between tRNA isoacceptors requires precise discrimination by the ribosome in order to maintain correct pairing between families of codons and their corresponding tRNAs in SCR applications. Hyperaccurate ribosomes (also known as error-restrictive ribosomes) that have a mutation in the S12 protein were uncovered decades ago due to their resistance to the antibiotic streptomycin[36]. These ribosomes show improved discrimination against near-cognate AA-tRNA competitors due to their enhanced proofreading during the accommodation of the EF-Tu-GTP-tRNA ternary complexes. (Fig. 2a)[37–40]. We wondered to what extent these ribosomes might improve codon orthogonality with cognate tRNAs through the narrowing of their codon reading ability. To test this hypothesis, we repeated the codon competitions with both wt tRNA and t7tRNA mixtures (Supplementary Fig. 2), replacing the wt ribosomes with mS12 ribosomes (Fig. 2b).

Hyperaccurate mS12 ribosomes led to improved codon orthogonality with the post-transcriptionally modified wt tRNA. (Fig. 2b and Supplementary Fig. 3). The greatest improvement was observed with the CUA and UUA codons, which were now predominantly read by LeuVAG and Leu)AA respectively. However, tRNA LeuVAG retained its ability to competitively read the CUG, CUU, and CUA codons (Fig. 2b), evidence that the mS12 ribosomes behave like wt with regards to the cmo5U34 modification. Interestingly, LeuCAG was capable of reading the CUA codon regardless of ribosome type (Figs. 1c, 2b).

Much to our delight, pairing the t7tRNA mixture with mS12 ribosomes lead to remarkable orthogonality across all six leucine codons (Fig. 2b and Supplementary Fig. 3). As expected based on previous work[22], the CUG, CUU, and UUG codons revealed readthrough only by one major tRNA: the Watson-Crick pairing tRNA, except for CUU which does not have a native tRNA with a fully complementary anticodon (Fig. 2b). Most notable was the change in competitive readthrough of the CUA codon which was now predominantly read by the Watson-Crick paired LeuUAG (Figs. 1c, 2b). The t7tRNA/mS12 ribosome pairing showed at least 76% competitive readthrough by a cognate, Watson-Crick pairing tRNA for all six leucine codons, limiting readthrough by wobble pairing tRNAs to no >9% (except the CUU codon, which does not have a native tRNA with a fully complementary anticodon) and

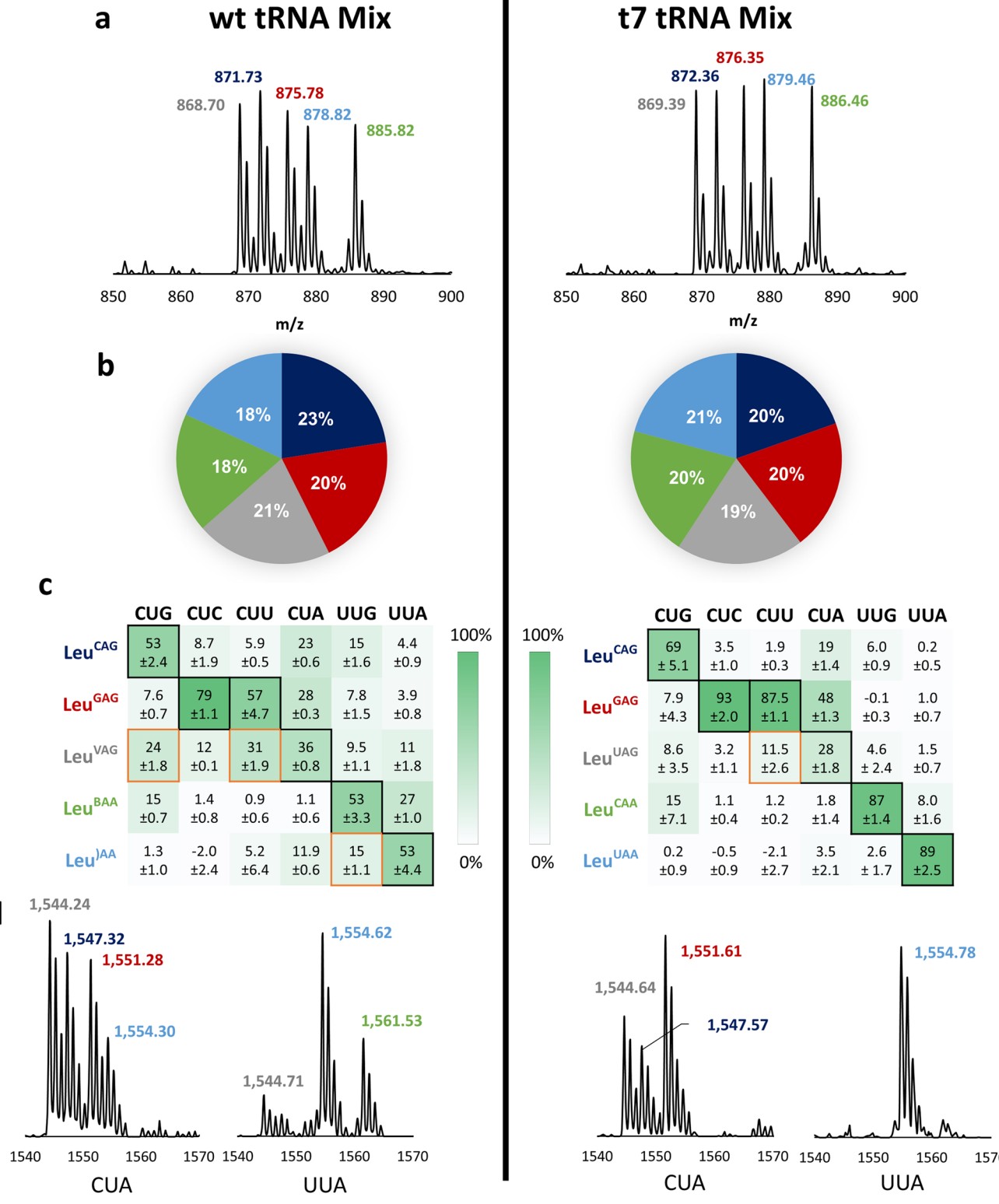

**Fig. 1 | Quantitative codon competition assay reveals codon readthrough patterns by wt and t7tRNA. a** All five tRNAs were individually aminoacylated with standard leucine, or one of four leucine isotopes, and supplemented into the PURE translation system at an equal concentration (5 µM) in the presence of one of 6 mRNAs containing a single Leu codon. **b** MALDI-MS analysis of the pre-charged tRNA mixtures reveals the stoichiometric balance of the mixtures, which is shown in the pie charts. **c** A heatmap illustrating the results of the codon competition translation assay using a wt tRNA mixture (left) and t7tRNA mixture (right). The expected cognate and wobble interactions between tRNA and codons are depicted with black borders for Watson-Crick pairing and gold borders for predicted wobble pairing. A green gradient is used to depict the percent of codon readthrough by a given tRNA, and the corresponding percentage value is shown inside the box. Anticodon post-transcriptional modifications on the wt tRNA are depicted with single letter codes: V = 5-oxyacetic acid uridine (cmo5U), B = 2′-O-methyl cytidine (Cm),) = 5-carboxymethylaminomethyl-2′-O-methyl uridine (cmnm5Um). **d** representative mass spectra of two codon competition experiments: one for the CUA codon and one for the UUA codon. A table of observed vs expected MS values is shown in Supplementary Table 2, and MALDI-MS data used to generate the heat maps is shown in Supplementary Fig. 2. Source data are provided as a Source Data file.

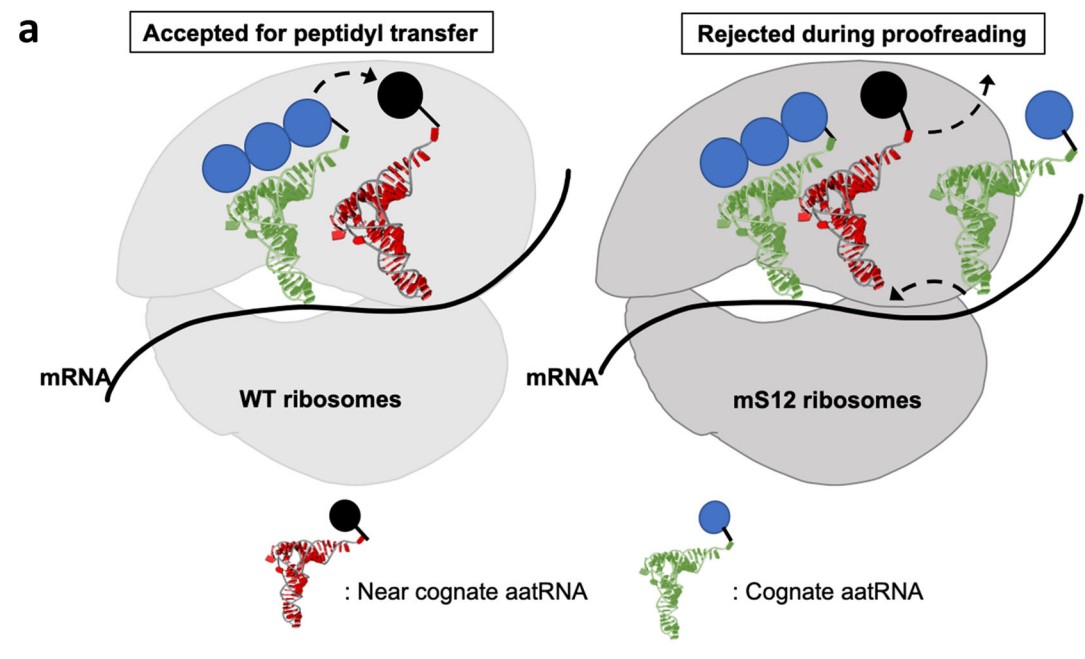

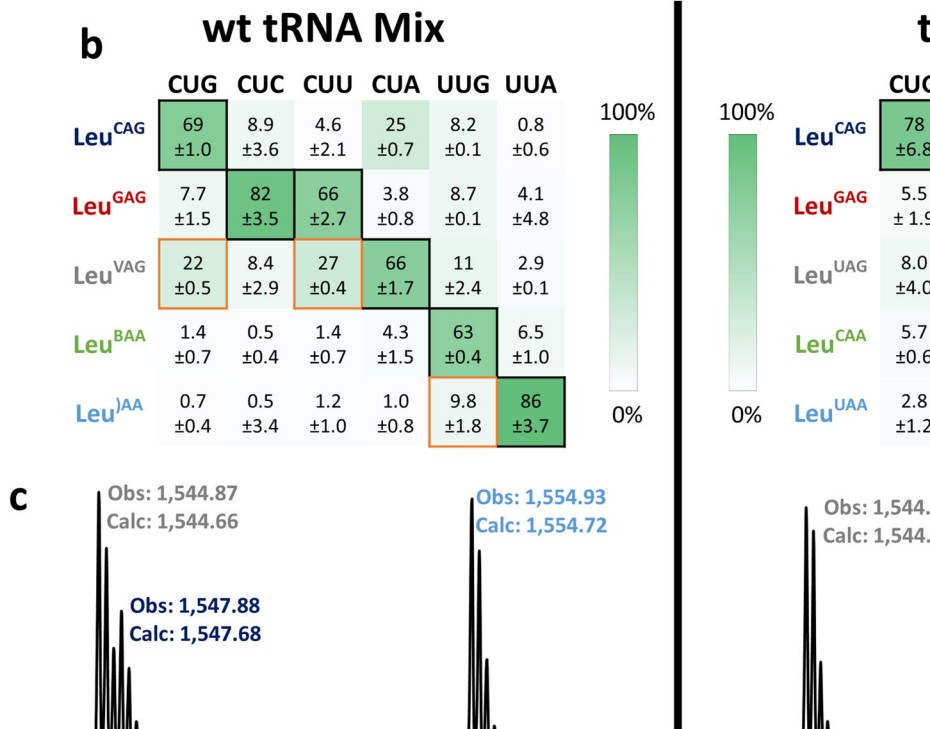

Fig. 2 | Codon competition assays show improved orthogonality with hyper-accurate ribosomes. a Translation using wild-type ribosomes can be prone to acceptance of near-cognate AA-tRNA, however, the hyperaccurate ribosome (mS12) rejects the near-cognate AA-tRNA, resulting in accurate peptide translation. b A heatmap illustrating the results of the codon competition translation assay using a wt tRNA mixture (left) and t7tRNA mixture (right). The expected cognate and wobble interactions between tRNA and codons are depicted with black borders for Watson-Crick pairing and gold borders for predicted wobble pairing. A green gradient is used to depict the percent of codon readthrough by a given tRNA, and the corresponding value is shown inside the box. Anticodon post-transcriptional

modifications on the wt tRNA are depicted with single letter codes: V = 5-oxyacetic acid uridine (cmo5U), B = 2'-O-methyl cytidine (Cm),) = 5-carbox-ymethylaminomethyl-2'-O-methyl uridine (cmnm5Um). c representative mass spectra of two codon competition experiments: one for the CUA codon and one for the UUA codon. Reactions were performed for thirty minutes at 37 °C with each AA-tRNA supplemented at 5 μM. A table of observed vs expected MS values is shown in Supplementary Table 3. MALDI-MS data showing the tRNA balance is shown in Supplementary Fig.3, and all of the MALDI-MS data used to make the heatmap is shown in Supplementary Fig. 4. Source data are provided as a Source Data file.

non-cognate tRNA pairings to no >7% (Fig. 2b). These results suggest the possibility of unprecedented reassignment of the six-fold degenerate leucine codon box, and perhaps the genetic code as a whole when using mS12 ribosomes and unmodified t7tRNAs.

While our t7tRNA mixture decoded the six leucine codons with greater orthogonality than the wt tRNA mixture, wt tRNAs are expected to have more favorable interactions with the ribosomal decoding center due to greater structural order provided by their modifications, as well as higher binding affinity to EF-Tu[29,41–43]. This inspired us to conduct our codon competition assay using a mixture of both wt and t7tRNA. For this mixture, we hypothesized that Leu$^{VAG}$ and Leu$^{)AA}$ could be replaced with t7 transcribed counterparts to reduce codon sharing, alongside wt Leu$^{CAG}$, Leu$^{GAG}$, and Leu$^{BAA}$. This assay, conducted with mS12 ribosomes, revealed reduced orthogonality relative to t7/mS12, similar to the pattern observed by the wt mixture alone except for reduced sharing of the CUG and CUU codons (Supplementary Fig. 4). These results highlight both the efficiency of the wt tRNAs and the challenge of mixing and matching wt and t7tRNAs for orthogonality.

### Application of the codon reading rules leads to extensive SCR of the sixfold degenerate leucine codon box

Having established codon readthrough patterns in a competitive format, we next sought to apply our high-performance t7tRNA/mS12 ribosome mixture to SCR. We designed a mRNA template with five leucine codons, choosing to omit CUU in favor of CUC (both are read by tRNA Leu$^{GAG}$). We pre-charged a balanced mixture of the five isotopologues onto t7tRNAs for supplementation into the PURE translation system (Fig. 3a). The SCR proof of concept was performed in triplicate, and in each case, MALDI-MS analysis of the translation results revealed a single major peak matching the expected mass of the desired peptide (Fig. 3b) and in alignment with the predicted isotope distribution (Fig. 3c) despite there being 126 possible combinations (9!/(4!5!)) of peptides of unique mass formed. These results validate that the orthogonality revealed for individual codons using t7tRNA paired with mS12 ribosomes can be applied to high-fidelity SCR, resulting in the most complete reassignment of a six-fold degenerate codon box described to date.

To explore the potential of the t7tRNA−mS12 ribosome pairing for SCR, we focused on maximal reassignment of the leucine codon box. We then applied our same pre-charging method to prepare the following ncAA-tRNA mixture: benzyl histidine−tRNA Leu$^{CAG}$, tert-butyl alanine−tRNA Leu$^{GAG}$, norvaline−tRNA Leu$^{UAG}$, tri-fluoro leucine−tRNA Leu$^{UAA}$, leucine−tRNA Leu$^{CAA}$ [34,44] (Fig. 3d). Translation in this context would require the incorporation of 4 consecutive ncAAs, which can be challenging[45]. Translation of this template with pre-charged t7tRNA and mS12 ribosomes yielded a pure mass spectrum profile, with a major peak corresponding to a peptide bearing leucine, tert-butyl alanine, norvaline, tri-fluoro leucine, and benzyl histidine (Fig. 3e). Benzyl histidine is a previously unreported substrate for a LeuRS mutant evolved to accept O-methyl tyrosine (Supplementary Fig. 5)[46]. To achieve efficient re-encoding with high fidelity, the concentration of pre-charged tRNAs had to be elevated to 15 μM, with BzHis-tRNA Leu$^{CAG}$ being added at 30 μM (Supplementary Fig. 6). We note here that these experiments were not the result of an extensive optimization process to determine which ncAA works best on which tRNA−the tRNA and ncAA pairings were made randomly. We, therefore, repeated these experiments, but swapped the tRNAs containing benzyl histidine and tri-fluoro Leu. While the major peak in these experiments was still the expected one, the orthogonality was not as clean, but could be improved through rational optimization of tRNA concentrations (Supplementary Fig. 7).

### Extensive reassignment of the valine codons

We wondered if the identified codon orthogonality with t7tRNAs and hyperaccurate ribosomes would apply to other degenerate codon boxes. The four-fold degenerate valine codon box is read by two tRNAs in *E. coli*, one bearing a GAC anticodon, and the other bearing cmo⁵U modified uridine (VAC). The codon reading patterns of the t7tRNAs were evaluated by Alexandrov[22], who found the tRNA lacking the cmo⁵U modification (tRNA$^{UAC}$) could still engage all 4 codons, making this a potentially difficult codon box to reassign effectively. We created two t7tRNAs (Supplementary Fig. 8), bearing the GAC and UAC anticodons, anticipating based on our results with the Leu codons that tRNA Val$^{GAC}$ would engage the GUU and GUC codons, the tRNA Val$^{UAC}$ would engage the GUA codon, and to some extent the GUG codon due to the lack of a competitor tRNA capable of readthrough. As expected, we saw strong orthogonality for the GUG, GUC, and GUA codons, and to a lesser extent the GUU codon (Fig. 4c, Supplementary Fig. 9). We also created a mutant tRNA$^{Val}$ with a CAC anticodon in an attempt to develop an orthogonal tRNA for the GUG codon; however, this tRNA showed promiscuous codon reading (Supplementary Fig. 10).

Our competition assays suggested the GUG, GUC, and GUA codons could be orthogonally read, prompting us to design a mRNA template bearing these codons to be decoded by valine−t7tRNA Val$^{UAC}$ and tert-butyl glycine−t7tRNA Val$^{GAC}$ (Fig. 4e). In this experimental design valine is doubly encoded by the GUA and GUG codons. MALDI-MS analysis of the translation results revealed a major peak matching the expected mass of the desired peptide (Fig. 4f) indicating that we were able to predictably reassign the valine codon box to two amino acids, using the information obtained from the codon competition experiments. This result is consistent with that of Suga who reported splitting up the GUG and GUC codons[15].

### Reassignment of the leucine and valine codon boxes to seven amino acids

Finally, we tested to what extent the competitive codon reading rules from both the Val and Leu codon boxes could be combined into a dramatically expanded code. We designed three mRNA templates that together encode for 9 of the 10 Val and Leu codons (Fig. 5a). We pre-charged each of the seven t7tRNAs using the encoding strategy shown in Fig. 5b or with standard valine and leucine and tested the mixtures in PURE translation reactions containing the mS12 ribosomes. We compared the yield of the translations and noted that the ncAA containing translations had lowered yields (20−30% of canonical) in line with expectations (Fig. 5c and Supplementary Fig. 11)[47]. Importantly, in each translation with ncAAs, the major peak was the peptide of correct mass (Fig. 5d and Supplementary Fig. 12). mRNAs 1 and 3 led to a single peptide with complete codon orthogonality, while mRNA 2 showed a couple of additional peaks resulting from a misreading of the CUG codon by norval-tRNA Leu$^{UAG}$ and tBuAla-tRNA Leu$^{GAG}$ instead of BzHis-tRNA Leu$^{CAG}$.

## Discussion

### Evaluation of competitive tRNA reading patterns for wild-type tRNAs

In this work, we have established the competitive nature of codon readthrough for the leucine and valine codon boxes, allowing a hierarchical ranking of tRNAs for each codon and a more complete picture of ribosomal discrimination of non and near-cognate tRNAs. In order to better visualize ribosomal discrimination patterns, in Fig. 6a we have re-visualized the heat maps from Fig. 1c (wt tRNA−wt ribosomes) and Fig. 2b (wt tRNA−mS12 ribosomes) to show the pairings between the third codon position and the tRNA wobble position (34th nucleotide). Using borders, we have separated the six codons into the CUN and UU(G/C) families, which are decoded by tRNAs containing a G$_{36}$ nucleotide and A$_{36}$ tRNA respectively. As expected, the A$_{36}$ tRNAs exhibited very little readthrough of the CUN codons, nor did the G$_{36}$ tRNAs prove efficient at reading the UU(G/C) codons, likely due to the ribosome's ability to enforce the W-C base pair at the first codon position.

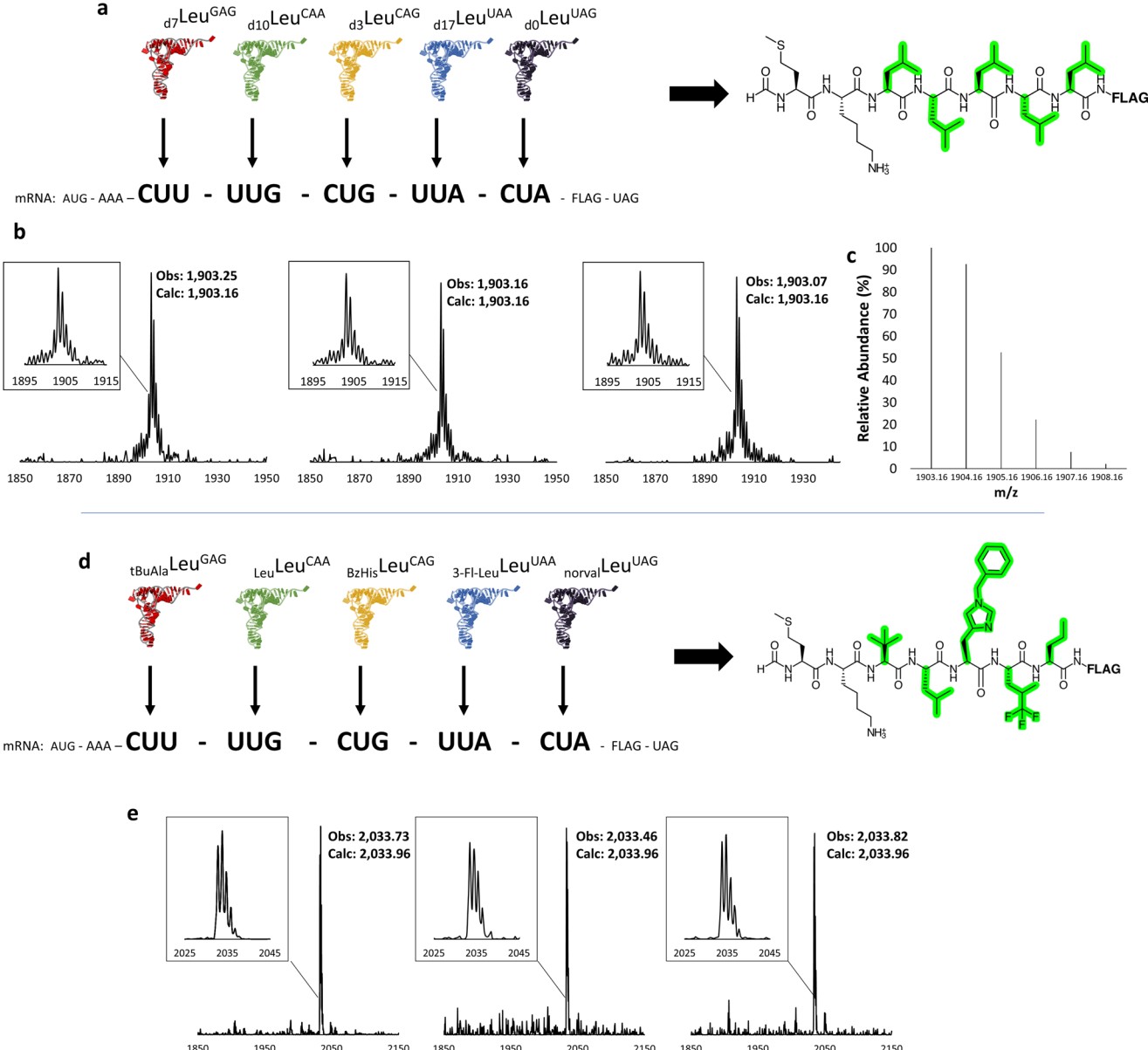

**Fig. 3 | Sense codon reassignment of the leucine codon box to encode 5 distinct amino acids. a** A single mRNA with five adjacent leucine codons is decoded in the PURE translation system by five pre-charged leucine−tRNA isoacceptors, each bearing a leucine isotope with a unique mass, to translate the shown peptide (leucine side chains are highlighted in green, deuterium not shown). S12 ribosomes and t7tRNA were used. **b** mass spectra of the translation results, performed in triplicate, with blown-up spectra included for analysis of peak distribution around the expected mass of 1903 Da. Translations were performed for 30 minutes at 37 °C, with all pre-charged AA-tRNAs at a final concentration of 5 μM. **c** Predicted isotope distribution for the in vitro translated peptide product (assuming 100% isotope substitution). **d** the same mRNA used in part a is decoded in the PURE translation system by five pre-charged leucine−tRNA isoacceptors, each bearing a different leucine analog (tBuAla tert-butyl alanine, BzHis benzyl histidine, 3-Fl-Leu trifluoroleucine, norval norvaline, and Leu leucine) in order to translate the shown peptide (leucine analog side chains are highlighted in green). S12 ribosomes and t7tRNA were used. **e** mass spectra of the translation results, performed in triplicate, with blown-up spectra included for analysis of peak distribution around the expected mass of 2033.95 Da. Translations were performed for 30 minutes at 37 °C, with each pre-charged AA-tRNA at a final concentration of 15 μM, except BzHis-Leu$^{CAG}$, which was added at 30 μM. Source data are provided as a Source Data file.

Our studies revealed that Watson-Crick and G:U pairing at the third codon position was preferred by the wt ribosomes (Fig. 6a), but that these ribosomes also accept C-A, G-A, Cm-A cmo⁵U-G, and cmo⁵U-U pairings. The cmo⁵U-G, cmo⁵U-U are known pairings that have been extensively studied[23,28,48]. Most surprising to us was the extent of competition at the CUA codon with tRNAs that have a C-A and G-A mismatch at the third position. If protonated, stable hydrogen bonding has been reported between C-A⁺ in both RNA and DNA, however, we have found only one instance of this pairing being reported as acceptable in translation[31]. Interestingly, the hyper-accurate ribosomes proved effective at discriminating against the G-A,

but not C-A, mismatch, eliminating the reading of the CUA codon by Leu$^{GAG}$ (Fig. 6). tRNAs that decode mixed codon boxes, such as UU (G/C), frequently feature anticodon modifications believed to narrow their codon recognition, making the observed Cm-A (and to a lesser extent cmnm⁵Um-G) pairings surprising, as these modifications are thought to restrict the codon reading of tRNA Leu$^{BAA}$ and tRNA Leu$^{)AA}$ [27]. Previous analysis of UU(G/C) readthrough by Leu$^{BAA}$ and Leu$^{)AA}$ in competition revealed readthrough of the UUA codon by Leu$^{BAA}$, although it was diminished when Leu$^{)AA}$ was involved as a competitor, and shared readthrough of UUG by both tRNAs, with an approximate 2:1 readthrough ratio in favor of Leu$^{BAA}$ [49]. These results

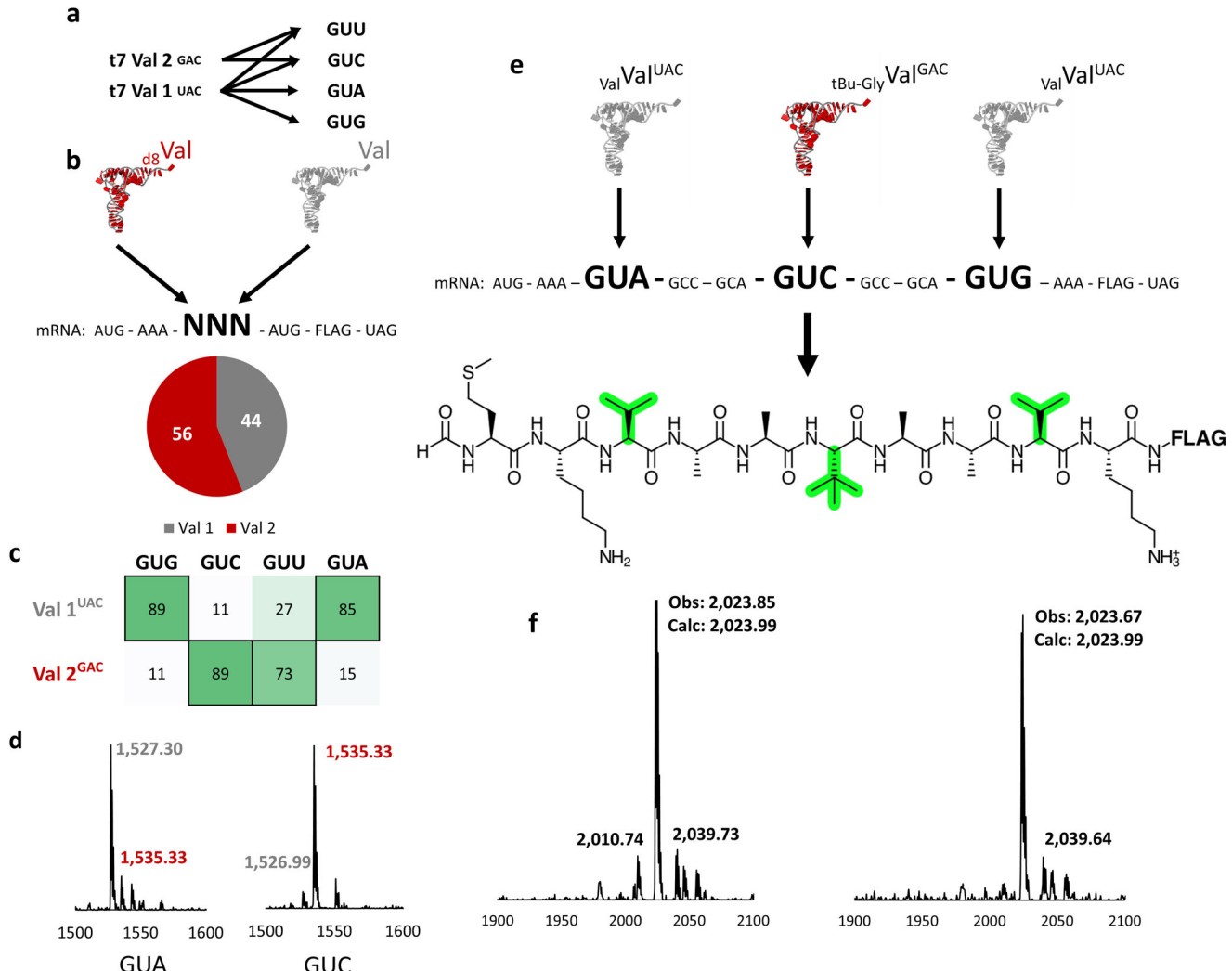

**Fig. 4 | Codon competition and sense codon reassignment of the four-fold degenerate valine codon box. a** Reported readthrough of each valine codon by t7tRNAs[22]. **b** MALDI-MS analysis of the pre-charged tRNA mixtures reveals the stoichiometric balance of the mixtures, which is shown in the pie chart. **c** Heatmap displaying the results of the codon competition experiments, where darker green corresponds with higher readthrough and the corresponding readthrough value numerically shown inside the box. The expected codon readthrough, based on patterns revealed by surveying the leucine codons, is depicted with a black box. A table of observed vs expected MS values is shown in SI Table 4. **d** Representative mass spectra of two codon competition experiments: one for the GUA codon and one for the GUC codon. All other mass spectra are shown in SI Fig. 5. **e** A single mRNA with three valine codons is decoded in the PURE translation system supplemented with two pre-charged valine tRNA isoacceptors, resulting in the translation of a peptide with valine and tert-butyl glycine (tBu-Gly highlighted in green). S12 ribosomes and t7tRNA were paired for these translation experiments, which were performed in duplicate. **f** MALDI-MS analysis of the translation results (expected mass of 2023.99 Da). Source data are provided as a Source Data file.

are mostly consistent with ours, except we show that Leu[JAA] remains a competitive presence when at equal concentrations with Leu[JAA] (Fig. 6a). The mS12 ribosomes were able to improve the orthogonality of these codons, with a significant reduction in readthrough of UUA by Leu[BAA] (Fig. 6a). Our codon competition assays with wt tRNAs reveal the challenging nature of ribosomal discrimination of modified tRNA, and present mS12 ribosomes as an attractive solution to these problems with improved discrimination against both non-cognate and near-cognate tRNAs.

### Improvements in orthogonality with mS12 and t7tRNAs
When combining mS12 ribosomes and t7tRNAs we saw dramatic improvements with codon orthogonality as shown in the heatmap in Fig. 6b. Most notable are the improvements in readthrough of the CUA codon, which showed a 52% increase in readthrough by cognate Leu[UAG]. We also observed reduced sharing of the CUG and CUU codons with a corresponding 24% increase in readthrough by their cognate tRNAs, and improved readthrough of the UUG and UUA

codons by their cognate tRNAs, each exhibiting a 37% and 29% increase in readthrough, respectively (Fig. 6b). Importantly, our valine codon competition experiments supported the trends observed with the leucine codons. cmo5U34 modified Val[VAC] is known to engage all four valine codons[23]. Using mS12 ribosomes and the t7 version, Val[UAC], lacking the cmo5U34 modification lost its ability to read the GUC and to a lesser extent, the GUU codon (Fig. 4c). While cmo5U34 deficient tRNA is not predicted to be able to read G-ending codons, our results showed that Val[UAC] retained its ability to dominantly read the GUG codon, likely due to the absence of a viable competitor tRNA[28]. Together, surveying of the leucine and valine codon boxes revealed that the t7tRNA−mS12 ribosome pairing enabled the primary engagement of a codon by a single tRNA for 9 of the 10 codons, with minor competitors reading the codons at most 9% (Figs. 2b, 4c). The remaining GUU codon was read in a 73:27 ratio (Fig. 4c). These results suggest that the discriminatory capabilities of the mS12 hyperaccurate ribosomes paired with diminished wobble readthrough by t7tRNA lacking the cmo5U34

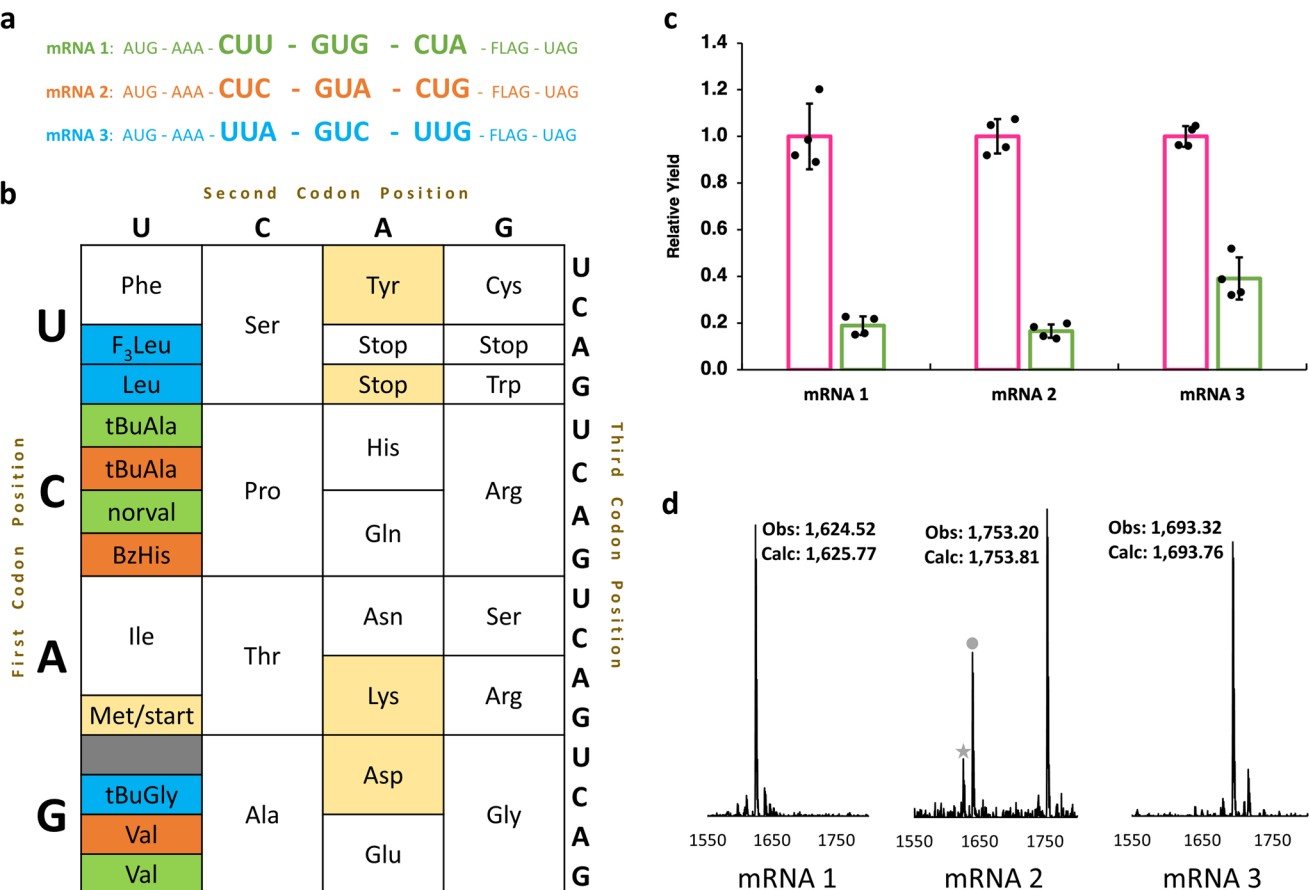

**Fig. 5 | Reassignment of the leucine and valine codon boxes to encode 7 distinct amino acids. a** The leucine and valine codons, with the exception of GUU (gray), were distributed across three mRNAs, each bearing two leucine codons flanking a single valine codon. **b** The reassigned genetic code. Tert-butyl alanine, norvaline, and tert-butyl glycine were encoded by mRNA 1 (green), tert-butyl alanine, benzyl histidine, and tert-butyl glycine were encoded by mRNA 2 (orange), and tri-fluoro leucine, leucine, and valine were encoded by mRNA 3 (blue). Beige codon boxes represent other amino acids encoded within the mRNAs, and white codon boxes were unused. **c** Peptide yield experiments were performed using pre-charged tRNA mixtures with canonical leucine/valine (gray) or using the ncAA mixture (black) shown in **b**. 15 μL translations were performed for 30 minutes, with the pre-charged valine tRNAs at 10 μM, BzHis-tRNA Leu$^{CAG}$ at 30 μM, and the remaining Leu tRNAs at

15 μM. Data represent the mean ± standard deviation of four independent experiments. **d** Representative mass spectra analyzing the translation results using each of the three mRNAs. The translations were performed following the same methods as the quantitative translations presented in **c**, except reactions were 30 μL. Observed masses are shown next to each major peak (expected masses for mRNA 1, 2, and 3 are 1625.77 Da, 1753.81 Da, and 1693.76 Da respectively). In the mRNA 2 mass spectra, a BzHis to norval misincorporated peptide peak is shown with a gray circle, and a peptide with BzHis to norval and a valine to tBu-Gly misincorporations is depicted with a gray star. The experiments were performed in triplicate, with the other mass spectra in SI Fig. 12. tBuAla tert-butyl alanine, BzHis benzyl histidine, 3-Fl-Leu leucine, norval norvaline, and Leu leucine, tBu-Gly tert-butyl glycine, Val valine. Source data are provided as a Source Data file.

anticodon modification can be used to overcome genetic code redundancy, indicating great potential for SCR.

The direct application of the isotopologue codon competition rules for SCR with ncAAs does require some optimization. The most problematic ncAA in our reassignment was benzyl histidine, which diverges in structure significantly from Leu. This is observed most extensively in norval-tRNA Leu$^{UAG}$ and tBuAla-tRNA Leu$^{GAG}$ misreading the CUG codon (Fig. 5)[29]. We suspect that a limiting factor here is the translocation of BzHis-tRNA Leu$^{CAG}$ to the ribosome by EF-Tu-GTP. The EF-Tu-AA-tRNA-GTP ternary complex derives its binding affinity from both the amino acid and the tRNA body[50–53], and this can affect the incorporation of ncAAs[54]. Neither leucine nor tRNA$^{Leu}$ have notably high affinities for EF-Tu[51], and the large size of BzHis is likely to weaken the interaction with EF-Tu[52,55]. This could be a contributing factor to the lower orthogonality when we charged BzHis onto tRNA Leu$^{UUA}$ (Supplementary Fig. 8). This tRNA is predicted to bind less strongly to EF-Tu based on its T-stem sequence (C$_{51}$-G$_{63}$ vs. G$_{51}$-C$_{63}$)[53,56]. Suga's recent report further highlights the need for uniform tuning of EF-Tu affinity for improved incorporation of difficult ncAAs[53]. This will be a further

avenue for fruitful study and may permit a rational approach for choosing the best tRNA-ncAA pairings.

We envision that the codon orthogonality achieved by pairing t7tRNA with mS12 ribosomes could be applied to many other codon boxes that are natively decoded by a cmo5U-modified tRNA. These include the serine, proline, threonine, and alanine codon boxes, all of which are at least four-fold degenerate, making them attractive targets for SCR. Combining the codons we successfully surveyed with the aforementioned codon families, a total of 28 codons could in principle be reassigned to encode for 20, rather than 6 amino acids. This type of expansion requires the ability to charge these tRNAs with ncAAs. Here we focused on using the AARS enzymes for this purpose. A reasonably diverse set of ncAAs can be attached onto tRNAs using this approach; however, most of these are near-neighbors to the canonical AAs[57]. A far greater diversity of ncAAs can be attached onto tRNAs in a tRNA agnostic manner using the Flexizyme enzyme system[58,59]. It should therefore be quite feasible to attach each of these tRNAs to different ncAAs, leading to a dramatically expanded code with exciting applications in the creation of diverse peptide libraries using mRNA display.

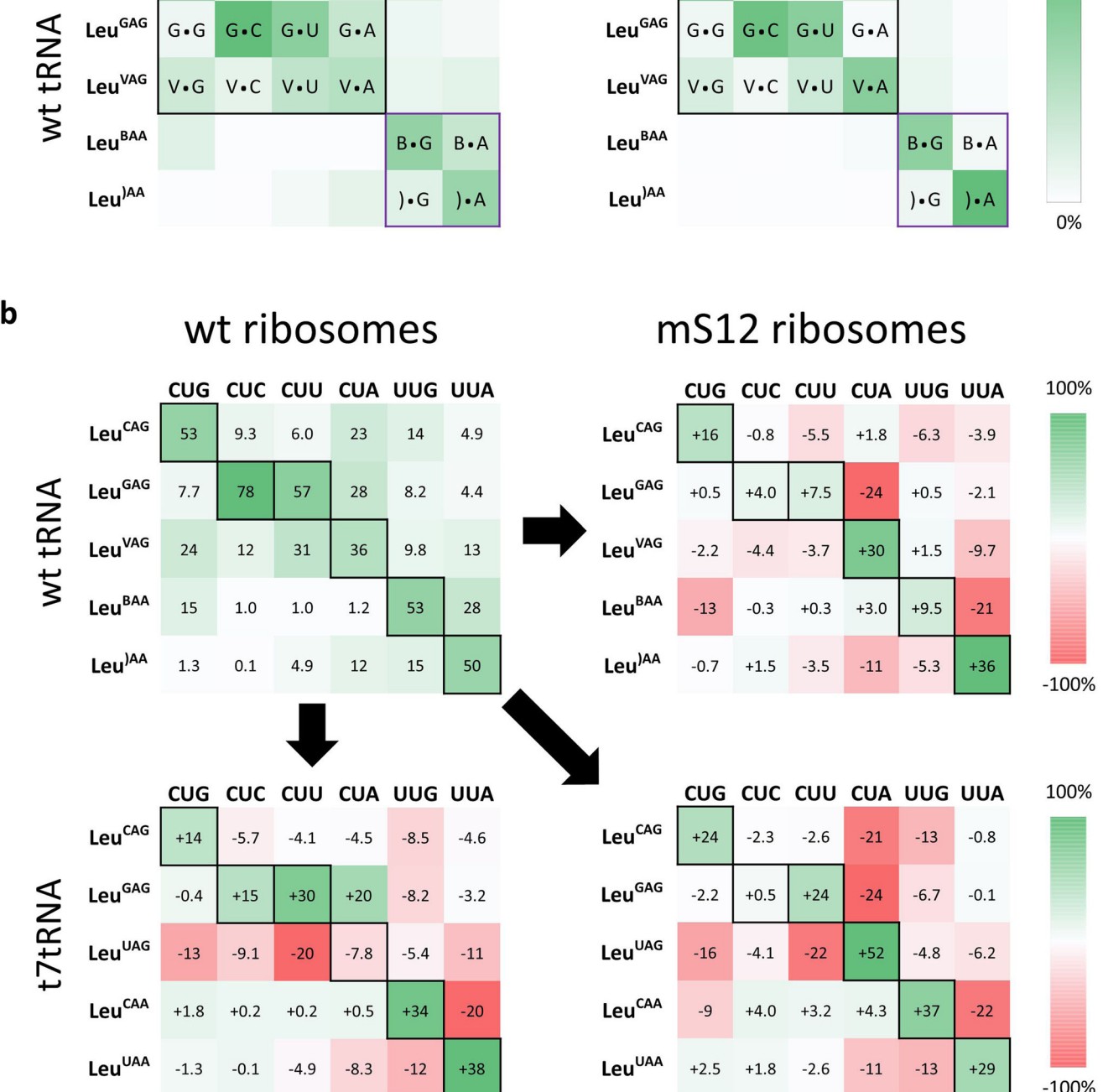

**Fig. 6 | Analysis of accepted base pairs at the third codon position by wt and S12 ribosomes and changes in codon readthrough intensity upon switching from a wt tRNA−wt ribosome to a t7tRNA−S12 ribosome pairing. a** Modified nucleotides are represented using single letter codes, where V represents cmo⁵U (5-oxyacetic acid uridine), B represents Cm (2′-O-methyl cytidine), and) represents cmnm5Um (5-carboxymethylaminomethyl-2′-O-methyl uridine) according to the Modomics web server abbreviations[60]. The tRNA base is listed first, and the mRNA second. Boxes with incorrect pairings at the first codon position were left unlabeled. Pairings between near-cognate tRNAs and codons from the unsplit CUN (N = G/C/U/A) and split UUN (N = G/A) codon box are encompassed with black and purple borders respectively. **b** A three-color heatmap is used to represent the quantitative changes in codon readthrough, where red corresponds to decreased readthrough, and green corresponds to increased readthrough. Black borders represent the desired codon readthrough for sense codon reassignment applications.

We describe a comprehensive surveying of codon readthrough in a competitive format, for both the leucine and valine codon boxes. Our assays reveal some unexpected codon sharing by tRNAs, which we were able to circumvent entirely by pairing t7tRNA with mS12 ribosomes. Using this combination, we were able to successfully reassign the leucine codon box to encode five distinct amino acids, including four ncAA. Furthermore, this pairing enabled the reassignment of the valine codon box to encode two distinct amino acids. Our work opens the doors to a dramatically reassigned genetic code with a host of applications in synthetic biology and drug discovery.

# Methods

## Reagents

*E. coli* total RNA was purchased from Sigma-Aldrich. Fluorous columns (Fluoro-Pak product # FP7210), phosphoramidite reagents, and fluorous CPG resin were acquired from BioSearch Technologies. The d3 (L-Leucine-5,5,5-d3 product # 486825), d10 (L-Leucine-2,3,3,4,5,5,5,5′,5′,5′-d10 product # 492949), and +17 (L-Leucine-13C6,15 N,2,3,3,4,5,5,5-D7 product # 749915-CONF) leucine isotopes were acquired from Sigma-Aldrich, and the d7 (L-Leucine-d7 (iso-propyl-d7 product # D-2661) leucine isotope was acquired from CDN Isotopes. Deuterated valine-$d_8$ was purchased from Cambridge Isotopes. Amino acids were dissolved in concentrations of 10–100 mM depending on solubility and the pH of the stock solution was adjusted to ~7.4 with 1 M KOH, monitoring with pH paper. Fluorous oligonucleotides used for the capture of the tRNAs are reported in ref. 33.

## Purification of tRNA leucines from *E. coli* total tRNA

The five wt-modified leucyl tRNA isoacceptors were purified to homogeneity using our previously reported fluorous capture method[33]. Lyophilized total RNA from *E. coli* MRE 600 (Roche, 10109550001) was re-dissolved in water to 100 mg/mL used as the tRNA stock for all capture experiments. Hybridization mixtures were prepared by mixing 600 µL of 5× hybridization buffer (1 M KCl, 0.5 mM EDTA, 250 mM HEPES, pH 7.5) with the appropriate fluorous-tagged oligo[33], adding 600 µL of total tRNA (100 mg/mL), and diluting with water to 3 mL. The amount of oligo used was matched to the predicted abundance of the target isoacceptor[35]. The mixture was then fully denatured on a thermocycler by heating at 90 °C for one minute, followed by 10 minutes of controlled heating at three degrees below the melting temperature (Tm). Fluoro-pak columns (BioSearch Technologies, FP7210) were coupled with a peristaltic pump, and the columns were pre-conditioned as follows: 8 mL of 100% acetonitrile (MeCN) was passed through the column, followed by 2 mL of triethyl ammonium acetate buffer (100 mM, pH 7.0) and finally 2 mL of loading buffer (1.71 M NaCl in 5% aqueous N,N-dimethylformamide). For the conditioning steps, the pump was set to a flow rate of 1 mL/minute. Pre-hybridized sample (see above) was then mixed 1:1 with loading buffer and passed through the column at a rate of 0.4 mL/min. After loading, the resin was subjected to a gradient wash of increasing stringency. The gradient was a mixture of loading buffer and wash buffer (10 mM TEAA in 10% aqueous MeCN), applied in 2 mL increments per column at a rate of 1 mL/minute in the following ratios of loading buffer to wash buffer: 85/15, 70/30, 55/45, 30/70, 15/85. The final wash was performed in triplicate.

To elute the tRNA, 2 mL of 100% wash buffer was added to the capped column and placed on a heat block at 85 °C for 3 minutes, and then immediately eluted. Samples were butanol concentrated from 2 mL to a volume suitable for ethanol precipitation in a 1 mL microcentrifuge tube. The recovered tRNA pellets were resuspended in water and stored at −80 °C.

## Equipment

Urea PAGE gels were imaged on a ChemiDoc MP Imaging System (BioRad) after staining with Sybr Green II. Sample absorbance was measured on a NanoDrop ND-1000 SpectroPhotometer. Mass spectrometry experiments were performed on a Voyager-DE Pro BioSpectrometry Workstation (Applied Biosystems) under reflectron positive mode.

## In vitro transcription of t7tRNA and mRNA

DNA templates encoding a 5′ T7 promoter sequence and a Shine Dalgarno sequence followed by a peptide coding sequence for in vitro transcription were prepared for mRNA using the polymerase chain reaction (PCR, oligonucleotide sequences are listed in SI Table 1). For tRNA this process was performed in the same manner, however, the Shine Dalgarno sequence and encoding sequence was replaced with the tRNA sequence (SI Table 2). PCR was performed in three steps: the first occurring between two oligos to establish the middle region of the template, the second involving the previously constructed template and a forward and reverse oligo, and the third involving the resulting extended template plus the same forward oligo with a new reverse oligo bearing a 2′-O-Methyl modification on the final guanosine in the sequence. PCR mixtures were prepared to 20 µL for the first step, 100 µL for the second step, and 1 mL for the third, each in water with a final oligonucleotide (Integrated DNA Technologies) concentration of 1 µM each, 1× of Q5 reaction buffer, 200 mM dNTPs, and 10 µL of Q5 DNA polymerase (New England BioLabs). For the second and third steps, 10 µL of the PCR product from the previous step was added. PCR was then carried out on a DNA Engine thermocycler (BioRad), beginning with an initial 1 minute denature step at 95 °C, followed by twenty cycles of the following: 98 °C for 20 seconds, oligo Tm for 40 seconds, 72 °C for 40 seconds. PCR products were purified via phenol/chloroform extraction followed by ethanol precipitation and resuspended to 100 µL in water. The PCR product was then in vitro transcribed overnight at 37 °C in a 1 mL solution with final concentrations of the following: tris pH 7.8, (40 mM), 0.1% TritonX-100, spermidine (2.5 mM), $MgCl_2$ (25 mM), DTT (10 mM), and 5 mM each of the nucleotide triphosphates (ATP, CTP, GTP, UTP), an additional 4 mM GTP, 0.2 U/µL ribosafe, 0.2 µM T7 polymerase, 0.001 mg/mL inorganic pyrophosphatase. Transcribed mRNAs and tRNAs were purified via electrophoresis. The target band was then excised from the gel, UV shadowed, and the sample was extracted from the gel via crush and soak (the excised gel was crushed in a tube, 8 mL of 0.3 M KCl was added, and the mixture was tumbled overnight at room temperature). The aqueous fraction was then collected from the mixture via syringe filtration, ethanol precipitated, and the final pellets were resuspended in water to a final concentration of ~100 µM.

## Expression of LeuRS mutant OMeYRS

The variant of EcLRS that has previously evolved to accept O-methyl-tyrosine[46] was used to encode benzyl histidine. This mutant contains the following mutations: M40L, L41E, T252A, Y499R, Y527A, H537G. The gene encoding this variant was PCR amplified using the oligos: GACGAGAGCAAAGAGAAGTATTACTGCC and CCCTGACACA GCAACTGTTTCGCTGGTTCGTC. A separate PCR reaction using the pET21a-LeuRS-His plasmid (Addgene #124113) was prepared using oligos GGCAGTAATACTTCTCTTTGCTCTCGTC and GACGAACCAGC-GAAACAGTTGCTGTGTCAGGG. These two fragments were subjected to Gibson Assembly (NEB) and the resulting new plasmid was transformed into BL21 (DE3) strain of *E. coli*. Two liters of LB broth were inoculated and incubated until exponential phase growth ($OD_{600} = 0.6$), and then expression was induced with IPTG to a final concentration of 0.1 mM. The culture was resuspended in 20 mL of Wash Buffer (50 mM $NaH_2PO_4$, 300 mM NaCl, 20 mM imidazole, 5 mM BME, pH = 8), then lysozyme was added to a final concentration of 1 mg/mL. The mixture was shaken for 1 hour, then lysed by sonication. Cell debris was pelleted by centrifugation and the lysate was removed. 2 mL of GoldBio Ni-NTA resin slurry was added to the lysate and the mixture was shaken at 4 °C for 1 hour. The bound resin was washed with 2 × 20 mL wash buffer, then eluted with 6 × 1 mL elution buffer (50 mM $NaH_2PO_4$, 300 mM NaCl, 250 mM imidazole, 5 mM BME, pH = 8). Elutions were analyzed by SDS-PAGE gel electrophoresis and then dialyzed overnight into 1 L of dialysis buffer (50 mM HEPES-KOH, 100 mM KCl, 10 mM $MgCl_2$, 7 mM BME, 30% glycerol, pH = 7.6) and stored at −80 °C.

## Urea PAGE analysis

tRNA and mRNA were analyzed on denaturing urea PAGE gels (12%, 1.0 mm), which were prepared in TBE buffer(1x) and pre-run at 200 V for at least 15 mins. 30 ng of sample, usually around 1 µL, was mixed

with 10 µL of urea (8 M) dye mixture (containing xylene cyanol and bromophenol blue), and diluted to 20 µL with water. The sample was heated at 90 °C for one minute before carefully loading onto the gel, which was run at 200 V for 45 minutes or until the xylene cyanol had migrated to the very bottom of the gel. The gel was then carefully removed and stained for 30 minutes with Sybr Green II (Molecular Probes) as specified by their user guide. Gels were imaged on a BioRad Gel Doc XR, and densitometric analysis was performed using the Image Lab software.

## Calculating tRNA concentration

Total sample absorbance was measured on a NanoDrop ND-1000 Spectrophotometer in nucleic acids mode. The A260 value was input into a web tool (http://biotools.nubic.northwestern.edu/OligoCalc.html) along with the sample sequence to determine the total concentration of the sample. PAGE analysis of purified wt leucyl tRNAs (but not t7tRNAs) sometimes revealed non-tRNA bands. In order to determine the true concentration of such samples, densitometric analysis was performed by determining the ratio of target band intensity to total lane intensity. This corrected value was multiplied by the total concentration to give the true concentration of the tRNA isoacceptor.

## tRNA pre-charging

Homogeneous tRNA samples were enzymatically aminoacylated with AA or ncAA according to a previously reported method and were analyzed by MALDI[33]. An individual tRNA was mixed with HEPES/KOH (30 mM pH = 7.4), $MgCl_2$ (15 mM), KCl (25 mM), ATP (6 mM), inorganic pyrophosphatase (0.001 mg/mL) (Sigma-Aldrich Cat#: I5907-1MG), a corresponding aminoacyl-tRNA synthetase (3 µM), a corresponding natural or unnatural amino acid (1 mM and 5 mM final, respectively), and bovine serum albumin (BSA, 0.24 mg/mL, previously dialyzed into deionized water) in a final volume of 50 µL. Reactions were also incubated with BME (1300 mM). The reaction was incubated for 30 min at 37 °C (120 min for unnatural amino acids) before quenching by adding 0.1 volumes of aq. NaOAc (3 M pH = 5.2). The RNA-containing aqueous phase was extracted by mixing it with a mixture of phenol/$CHCl_3$/isoamyl alcohol (25/24/1) (unbuffered) and subsequently vortexed and centrifuged to separate layers. The aqueous layer was removed and transferred to a clean vial and mixed with an equal volume of $CHCl_3$ and vortexed and centrifuged again. The aqueous layer was then removed into a clean vial and mixed with three volumes of ice-cold ethanol (100%). The vial was incubated at −20 °C for 20 min and centrifuged for 20 min at 4 °C at $17,000 \times g$. The pellet was twice washed with 500 µL of 70 % Ethanol and then once more with 500 µL of 100 % ethanol to remove salts. Finally, the pellet was left to air dry and resuspended in NaOAc (12.5 µL, 100 mM pH = 5.0). Reductive amination was carried out by mixing 6.25 µL of the previous tRNA-aa preparation with 3.75 µL of $dH_2O$, (4 formylphenoxypropyl)triphenyl-phosphonium bromide in MeOH (12.5 µL, 63 mM) and fresh $NaBH_3CN$ dissolved in 50 mM NaOAc pH = 5.0 (2.5 µL, 200 mM). The reaction was incubated at 37 °C on tumbler for 2 hours before quenching with 0.1 volume of $NH_4OAc$ (4.4 M pH = 5.0). The reaction product was recovered through ethanol precipitation and the resulting pellet was resuspended in of $NH_4OAc$ (2.25 µL, 200 mM pH = 5.0). 0.25 µL of Nuclease P1 (1 IU/µL in 200 mM $NH_4OAc$ pH = 5.0) (Wako Cat#: 145-08221) was added and incubated at rt for 20 min. After incubation, the reaction mixture was quenched on ice and 1 µL was mixed with 9 µL of MALDI matrix α-cyano-4-hydroxycinnamic acid (CHCA) 10 mg/mL in MeCN: 2 % TFA (1:1) and spotted onto the MALDI plate for further analysis.

## Codon competition assay and SCR

All experiments were performed with a customized version of the PURE cell-free translation system[14]. All translations, unless otherwise specified, were performed on a 30 µL scale for 30 minutes at 37 °C. All amino acids and aminoacyl-tRNA synthetases (aaRS) required to decode the mRNA were added to the translation with the exception of the leucine/LRS pair. The final concentration of enzymes in the translation assay was as follows: EF-Tu (10 µM), EF-Ts (8 µM), IF-1 (2.7 µM), IF-2 (0.4 µM), IF-3 (1.5 µM), EF-G (0.52 µM), RF-1 (0.3 µM), RRF (0.5 µM), RF-3 (0.17 µM), ribosomes (1.2 µM), inorganic pyrophosphatase (0.1 µM), creatine kinase (4 µg/mL), nucleoside PP kinase (0.5 mg/mL), myokinase (4 µg/mL), and the aaRSs (0.1–1.0 µM). Pre-charged tRNAs were added to a final concentration of 5 µM unless otherwise specified. Translation reactions were initiated by adding the mRNA to a final concentration of 1 µM. After 30 min, the translation reaction was quenched with 90 µL of 1× TBS buffer (50 mM Tris-HCl, 300 mM NaCl, pH 8.0) and added to 10 µL of anti-FLAG M2 affinity resin (Sigma-Aldrich) in a nylon filter tube (VWR, product # 82031-358) and bound on a tumbler for 1 h. The resin was washed three times with TBS buffer (500 µL per wash) followed by elution in 50 µL of 1% aqueous TFA (250 µL TFA diluted to 25 mL in water). The resulting peptides were desalted via zip-tipping according to the manufacturer's protocol, eluted with CHCA matrix (recrystallized CHCA was resuspended to a final concentration of 10 mg/mL in 1:1 acetonitrile, 0.2% aqueous TFA), and analyzed via MALDI-MS.

Codon competition experiments were quantitatively analyzed by using the Voyager-DE Pro MALDI-MS ASCII data files exported to Excel (LTSC Professional Plus 2021) to compare peak intensities of peptides bearing different isotopologues. Codon competition experiments were performed in triplicate and peptide peak intensities were averaged together. The percent readthrough of a codon by a given tRNA was found by dividing the average total peak intensity for that peak by the average total peak intensity for all of the peaks (five peaks for the leucine competitions, two for valine). In cases where the mass difference between peptides was within four Daltons (i.e., leucine and d3-leucine, d3-leucine and d7-leucine, d7-leucine, and d10-leucine), the natural carbon isotope pattern was calculated and subtracted from the total intensity of the overlapped peptide, to determine the true intensity.

## Statistics and reproducibility

Quantitative MALDI experiments were performed in triplicate, and the numbers reported are the means ± the standard deviations. We did not attempt to replicate these experiments outside of the replicates included as data in the figures. The MALDI-MS assay for the AA-tRNA served as an internal control to ensure the pre-charged tRNAs were added at the same concentration. MALDI spots that gave no signal or very weak signal to noise were excluded. No statistical method was used to predetermine the sample size. The investigators were not blinded to allocation during experiments and outcome assessment.

## Reporting summary

Further information on research design is available in the Nature Portfolio Reporting Summary linked to this article.

## Data availability

The MALDI-MS spectra used to derive readthrough percentages are included in the Supplementary Figures. Source data is included in the Supplementary Information/Source Data file. Source data are provided with this paper.

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

## Acknowledgements

The authors would like to acknowledge the support of the NIH (R01GM143396 to M.C.T.H. and T.A.C.) for this work.

## Author contributions

Conceptualization: Cl.A.M., M.C.T.H., methodology: Cl.A.M., M.C.T.H., formal analysis: Cl.A.M., B.S., Ch.A.M., investigation: Cl.A.M., B.S., Ch.A.M., A.K.H, M.C.T.H., resources: A.K.H., T.A.C., writing-original draft: Cl.A.M., writing—review & editing: Cl.A.M., T.A.C., M.C.T.H., visualization: Cl.A.M., B.S., supervision: T.A.C., M.C.T.H., project administration: M.C.T.H., funding acquisition: T.A.C., M.C.T.H.

## Competing interests

Provisional patent applications by Virginia Commonwealth University covering the fluorous capture strategy (#22-018 F, inventors M.C.T. Hartman and C.A.L. McFeely) and the use of hyperaccurate ribosomes to expand the genetic code (#22-033 F, inventors M.C.T. Hartman, C.A.L. McFeely, and B. Shakya) have been submitted. C.A. Makovsky, A.K. Haney, and T.A. Cropp declare no competing interests.
