## [Peer Review File · Nature Communications]

REVIEWER COMMENTS

Reviewer #1 (Remarks to the Author):

Sense codon reassignment (SCR) enables ribosomal incorporation of non-canonical amino acids (ncAAs) into peptides/proteins, where ncAAs are assigned at any of the 61 sense codons in place of cAAs. Thus, the number of available amino acids should be generally no more than 20 in total. To overcome this limitation, the authors aimed at breaking the degeneracy of sense codons. Here, Leu (six-fold degenerate) and Val (four-fold degenerate) codon boxes could be successfully divided into five and two amino acids, respectively. The authors also showed that the use of T7-transcribed tRNA (t7tRNA) and hyperaccurate ribosome (mS12) could improve the accuracy of reassignment. I think these results open the door for extensive genetic code manipulation, enabling incorporation of many ncAAs. Therefore, I would recommend this manuscript for publication in Nature Communications if the following minor issues are addressed.

1. Page 5 “Of the three leucine codons reported to be “shared” between two tRNA isoacceptors (36), ...”

It is unclear which codons are discussed here.

2. Page 5 “In general, the wt tRNA mixture revealed significant codon overlap; an unattractive outcome for SCR applications.”

This result should not be “general”, but specific to the tRNAs that decode Leu codons. Use of other tRNAs bearing different types of nucleotide modification would give different results.

3. Page 8 “Extensive reassignment of the valine codons”

Compared to the reassignment of the Leu codon box, novelty of this part is minimum because only two amino acids could be assigned. Division of Val codon box into two using T7-transcribed tRNA has been previously demonstrated in ref (15), which should be referred to here.

4. In the figures that show mass spectra, both calculated and observed m/z values should be indicated to improve readability.

Reviewer #2 (Remarks to the Author):

McFeely et al. describe the development of a mass spectrometric assay using isotopically labeled amino acids to quantify codon use and reassignment. This was an elegant study which compared different variables including ribosomes, t7 tRNA, and modified non-canonical amino acids. Through a series of experiments, the authors show the power of their MS assay system for a quick and quantitative readout of codon usage. I had no major concerns for the MS experimental design, the authors take into account isotopic overlap in quantification of peptide peaks. I think it may be helpful in their first proof of concept experiment in Fig 3 to show that they indeed definitively get a single product (see below).

I recommend publication of this manuscript in Nature Communications after minor revisions.

Minor comments.

What is PURE? It's not defined in first use on Pg 3

Fig 1c: It is not clear what information is represented in the heat maps. They appear to be percentages but it should be stated more clearly in the legend.

Pg 7 and Fig 3b: The sense codon reassignment experiment in Fig 3 is elegant as it is a quick way to see if there is any nonspecific incorporation from one of the codons. The peptide peak looks a bit noisier in Fig 3b compared to Fig 3d, I think the authors should calculate the theoretical isotopic distribution as a means to show that their product is pure. Alternatively, they might show (either in a figure or a table) the theoretical distribution of masses/peaks if codon reassignment was unsuccessful/incomplete.

Fig 5d: There are some smaller peaks to the left and right of the major peaks in mRNA 1, 2, 3. It's not easy to tell but are they salt adducts? Are they assignable to any of the other tRNA-AA?

Reviewer #3 (Remarks to the Author):

Enthusiasm is high for this manuscript. The authors make a convincing argument that the best strategy for GCE may be breaking the degeneracy of the sense codons. They demonstrate that they can do this using hyperaccurate mS12 ribosomes at Leu and Val codons. The experiments in the manuscript are logical. In Figure 1 they evaluate wt and run-off tRNAs aminoacylated with all five isotope labeled Leus for translation of a single Leu codon. In Figure 2 they carry out the same experiment with hyperaccurate mS12 ribosomes, finding that the run-off tRNAs decode the six Leu codons with high degeneracy. In Figure 3 they show they can build on this experiment engineering a 5-Leu codon template and translating it either with 5 Leu with different isotope labels of 5 different unnatural amino acids. Figure 4 is along the same lines but does not go as far as with Leu (check the amino acid stereochemistry in this

Figure). In Figure 5 they encode 7 unnatural amino acids using a combination of Leu and Val codons. This system can be immediately employed for encoding unnatural amino acid oligomers using the Pure System. The idea is distinct from the work of others breaking sense codon degeneracy. It will be interesting to see how the results extend with S30 extracts and in vivo with stapled ribosomes.

On a more technical note, with reassignment of codons to synthesize unnatural oligomers, it is very important to confirm both the aa-tRNA reactants and the oligomer products. It is recommended that the authors confirm the structure of the aa-tRNA reactants. Also, in addition to mass spec, the oligomer products can be confirmed by radiolabeling at the N- and C-terminal positions and swapping amino acids internal in the oligomer sequence and carrying out co-migration analysis by HPLC.

Finally, the authors are missing some key references, e.g. manuscripts by Anthony Forster on sense codon reassignment.

Reviewer #4 (Remarks to the Author):

This manuscript applied a mass spec-based, "heavy/light"-multiplexing method extensively in order to delineate the interactions between individual tRNAs and codon boxes that underlie the same degenerate codes for leucine and valine, in the context of cell-free translation. While this principal technology is generally sound, there are a few points that will strengthen the quality of this manuscript and its claims:

- MALDI is known to have a poor performance in quantitative applications. For the heatmaps in Fig 1C, Fig 2B, Fig 4C, and Fig 6, these numbers are ratios calculated from comparing different mass peaks and processed to sum up to 100% per column, and they are further compared across columns in an overview to draw important conclusions of orthogonality. What is the typical range of % error in each cell value?
- In Fig 3C and 3D, the isotopic distributions of all three triplicates are not the same, also the mass error of the observed peak at 2033.37 is rather high (285 ppm). In Fig 5D, the difference between the observed vs. calculated mass of the mRNA1-translated peptide is also high (768ppm). It will be more convincing to apply the high-resolution mass spec to confirm the mass accuracy and rule out any other potential byproducts that have a similar mass.
- T7 tRNA was chosen over wt tRNA for reprogramming degenerate codons because of its observed better orthogonality. One concern with the T7 tRNA is the resultant decreased yield of the in vitro translated

peptides, which may still contain byproduct peaks with quantities that are close to the detection limit and affect the orthogonality assessment. Could you comment on the peptide yield (e.g. Fig 3 and Fig 5) and the detection limit of your MALDI method?

- This is an intensively studied field filled with controversial observations. In Suga's paper (Nat Chem. 2016 Apr;8(4):317-25; Fig 3), his group showed that tRNA-Val-CAC can specifically target GUG codon but not GUC. However, the current manuscript showed a significant crosstalk between tRNA-Val-CAC and the GUC codon (Omer-Thr readthrough, Fig S9). Could you address these inconsistent findings?

- To assess the freedom and establish the generality of assigning different non-natural amino acids to the chosen split codons, could you compare different permutations of the AA- tRNA pairing in Fig 3C and Fig 5B? For example, BzHis is the most "challenging" amino acid among all tested. Could it be only mapped to the CUG codon cleanly or does it get incorporated to other degenerate codons for Leucine equally well?

- Consider adding a paragraph to discuss the scope and/or the limitation of the non-natural amino acids that can be reprogrammed using this synthetase pre-charged method, in comparison to the wider range of monomers that can be utilized via flexizyme charging.

I would recommend a publication of the revised manuscript that has thoroughly addressed and incorporated the above comments.

We are grateful to the reviewers for their careful and thorough comments on our draft. Our responses to these critiques are shown below point-by-point in blue.

Reviewer #1 (Remarks to the Author):

Sense codon reassignment (SCR) enables ribosomal incorporation of non-canonical amino acids (ncAAs) into peptides/proteins, where ncAAs are assigned at any of the 61 sense codons in place of cAAs. Thus, the number of available amino acids should be generally no more than 20 in total. To overcome this limitation, the authors aimed at breaking the degeneracy of sense codons. Here, Leu (six-fold degenerate) and Val (four-fold degenerate) codon boxes could be successfully divided into five and two amino acids, respectively. The authors also showed that the use of T7-transcribed tRNA (t7tRNA) and hyperaccurate ribosome (mS12) could improve the accuracy of reassignment. I think these results open the door for extensive genetic code manipulation, enabling incorporation of many ncAAs. Therefore, I would recommend this manuscript for publication in Nature Communications if the following minor issues are addressed.

1. Page 5 “Of the three leucine codons reported to be “shared” between two tRNA isoacceptors (36), ...”

It is unclear which codons are discussed here.

We have added a clause specifying the three codons.

2. Page 5 “In general, the wt tRNA mixture revealed significant codon overlap; an unattractive outcome for SCR applications.”

This result should not be “general”, but specific to the tRNAs that decode Leu codons. Use of other tRNAs bearing different types of nucleotide modification would give different results.

We have added “for the leucine codon box” to the end of the sentence to narrow this claim.

3. Page 8 “Extensive reassignment of the valine codons”

Compared to the reassignment of the Leu codon box, novelty of this part is minimum because only two amino acids could be assigned. Division of Val codon box into two using T7-transcribed tRNA has been previously demonstrated in ref (15), which should be referred to here.

We have added a sentence: “This result is consistent with that of Suga who reported splitting up the GUG and GUC codons,” to this section.

4. In the figures that show mass spectra, both calculated and observed m/z values should be indicated to improve readability.

Thanks for this helpful feedback. We have added calculated and observed values to most MALDI-MS spectra in the main figures. In certain cases where there are multiple peaks observed (e.g. Fig 1d), we have just shown the observed peaks, since adding calculated mass peaks made these spectra difficult to read due to lack of space.

Reviewer #2 (Remarks to the Author):

McFeely et al. describe the development of a mass spectrometric assay using isotopically labeled amino acids to quantify codon use and reassignment. This was an elegant study which compared different variables including ribosomes, t7 tRNA, and modified non-canonical amino acids. Through a series of experiments, the authors show the power of their MS assay system for a quick and quantitative readout of codon usage. I had no major concerns for the MS experimental design, the authors take into account isotopic overlap in quantification of peptide peaks. I think it may be helpful in their first proof of concept experiment in Fig 3 to show that they indeed definitively get a single product (see below).

I recommend publication of this manuscript in Nature Communications after minor revisions.

Minor comments.

What is PURE? It's not defined in first use on Pg 3

PURE stands for "protein synthesis using recombinant elements" which is now defined in the manuscript.

Fig 1c: It is not clear what information is represented in the heat maps. They appear to be percentages but it should be stated more clearly in the legend.

Yes, these are percentages, and we have now added a gradient bar to each of these figures for clarity. A sentence in the legend now reads: "A green gradient is used to depict the percent of codon readthrough by a given tRNA, and the corresponding percentage value is shown inside the box."

Pg 7 and Fig 3b: The sense codon reassignment experiment in Fig 3 is elegant as it is a quick way to see if there is any nonspecific incorporation from one of the codons. The peptide peak looks a bit noisier in Fig 3b compared to Fig 3d, I think the authors should calculate the theoretical isotopic distribution as a means to show that their product is pure. Alternatively, they might show (either in a figure or a table) the theoretical distribution of masses/peaks if codon reassignment was unsuccessful/incomplete.

We have added an inset to Fig. 3b, which shows the theoretical distribution of peaks to Figure 3c. The expected isotope peak ratios are quite similar to the calculated ones. Potential peaks resulting from single misincorporations are shown below—all of these peaks would be at least 3 Da off from the major peak—these peaks are minimal in the observed spectra.

		Leu	Leu +3	Leu +7	Leu +10	Leu +17
	mass	113	116	120	123	130
Leu	113	0	-3	-7	-10	-17
Leu +3	116	3	0	-4	-7	-14
Leu +7	120	7	4	0	-3	-10
Leu +10	123	10	7	3	0	-7
Leu +17	130	17	14	10	7	0

Fig 5d: There are some smaller peaks to the left and right of the major peaks in mRNA 1, 2, 3. It's not easy to tell but are they salt adducts? Are they assignable to any of the other tRNA-AA?

We have looked carefully at the largest of these minor peaks, which are most significant in mRNA 1 and mRNA 3. In mRNA 1 (left side), we can assign two of the peaks, which have +14 and -14 masses. For mRNA 3 One of the peaks is a sodium adduct, and the other is -14. There are multiple possible substitutions that can lead to these misincorporations as shown in the misincorporation table we show here below the figure. Based on codon reading preferences, we think it is most likely that mRNA 1 is showing minor Val → tBuGly (+14) and tBuAla→Leu (-14) substitutions, and mRNA 3 is showing minor tBuGly →Val (-14) substitutions. We still point out that these breakdowns in orthogonality are very minor and don't change the overall impressive codon reading orthogonality when pairing t7 tRNAs and hyperaccurate ribosomes.

		Leu	norval	BzHis	f3Leu	tbu-ala	tbu gly	Val
	mass	113	99	227	167	127	113	99
Leu	113	0	14	-114	-54	-14	0	14
norval	99	-14	0	-128	-68	-28	-14	0
BzHis	227	114	128	0	60	100	114	128
f3Leu	167	54	68	-60	0	40	54	68
tbu-ala	127	14	28	-100	-40	0	14	28
tbu gly	113	0	14	-114	-54	-14	0	14
Val	99	-14	0	-128	-68	-28	-14	0

Reviewer #3 (Remarks to the Author):

Enthusiasm is high for this manuscript. The authors make a convincing argument that the best strategy for GCE may be breaking the degeneracy of the sense codons. They demonstrate that they can do this using hyperaccurate mS12 ribosomes at Leu and Val codons. The experiments in the manuscript are logical. In Figure 1 they evaluate wt and run-off tRNAs aminoacylated with all five isotope labeled Leus for translation of a single Leu codon. In Figure 2 they carry out the same experiment with hyperaccurate mS12 ribosomes, finding that the run-off tRNAs decode the six Leu codons with high degeneracy. In Figure 3 they show they can build on this experiment engineering a 5-Leu codon template and translating it either with 5 Leu with different isotope labels of 5 different unnatural amino acids. Figure 4 is along the same lines but does not go as far as with Leu (check the amino acid stereochemistry in this Figure). In Figure 5 they encode 7 unnatural amino acids using a combination of Leu and Val codons. This system can be immediately employed for encoding unnatural amino acid oligomers using the Pure System. The idea is distinct from the work of others breaking sense codon degeneracy. It will be interesting to see how the results extend with S30 extracts and in vivo with stapled ribosomes.

On a more technical note, with reassignment of codons to synthesize unnatural oligomers, it is very important to confirm both the aa-tRNA reactants and the oligomer products. It is recommended that the authors confirm the structure of the aa-tRNA reactants. Also, in addition to mass spec, the oligomer products can be confirmed by radiolabeling at the N- and C-terminal positions and swapping amino acids internal in the oligomer sequence and carrying out co-migration analysis by HPLC.

We do confirm the AA-tRNA species used in our translation reactions using a MALDI-AARS assay. Examples of the results of this assay are shown in Fig. 1d, Fig. S3a, and Fig. S6. The assay works by derivatizing the primary amino group of the amino acid on the aminoacyl-tRNA using reductive amination with a phosphonium-tagged benzaldehyde. The tRNA is then digested with nuclease P1 and the resulting derivatized AA-AMP is analyzed by MALDI. As a part of our workflow for this project, each AA-tRNA was subjected to this assay to confirm efficient charging.

We're having a hard time envisioning the experiment suggested for confirming the oligomeric sequence. We have carefully characterized our peptide products by MALDI-MS, which is the standard technique for studying in vitro translated peptides and has been used in dozens of ncAA introduction papers, including those by our lab.

Finally, the authors are missing some key references, e.g. manuscripts by Anthony Forster on sense codon reassignment.

We agree that we have left out certain Forster papers. In this revised manuscript, we have added the following references:

Ref 45: *NAR* 2009, 11:3747-3755

Ref 55: *RNA* 2014, 20: 632-643

Ref 56: *Curr. Opin. Chem. Biol.* 2018, 46:180–187

Reviewer #4 (Remarks to the Author):

This manuscript applied a mass spec-based, "heavy/light"-multiplexing method extensively in order to delineate the interactions between individual tRNAs and codon boxes that underlie the same degenerate codes for leucine and valine, in the context of cell-free translation. While this principal technology is generally sound, there are a few points that will strengthen the quality of this manuscript and its claims:

- MALDI is known to have a poor performance in quantitative applications. For the heatmaps in Fig 1C, Fig 2B, Fig 4C, and Fig 6, these numbers are ratios calculated from comparing different mass peaks and processed to sum up to 100% per column, and they are further compared across columns in an overview to draw important conclusions of orthogonality. What is the typical range of % error in each cell value?

The reviewer is correct that the heat maps should be read down the column, as the experiments were performed with a mixture of the tRNAs and a single codon. In each of the figures containing heat maps, we have now added the standard deviation errors and a heat map legends.

- In Fig 3C and 3D, the isotopic distributions of all three triplicates are not the same, also the mass error of the observed peak at 2033.37 is rather high (285 ppm). In Fig 5D, the difference between the observed vs. calculated mass of the mRNA1-translated peptide is also high (768ppm). It will be more convincing to apply the high-resolution mass spec to confirm the mass accuracy and rule out any other potential byproducts that have a similar mass.

The changes in isotopic ratios between the replicates in 3C are quite small, and are consistent with what we get on replicate data from our MALDITOF-MS. We also would like to point out that we are being more thorough here than most other papers that investigate in vitro translation data using MALDI that give a single, representative spectra.

The ncAAs we used in Figure 3D were chosen specifically to avoid ambiguity in mass. The minimum difference in these ncAAs from one another is 14 Da. We've shown the single substitution masses in a misincorporation table below. Any single misincorporation would be at least 14 Da different in mass.

		Leu	norval	BzHis	f3Leu	tbu-ala
	mass	113	99	227	167	127
Leu	113	0	14	-114	-54	-14
norval	99	-14	0	-128	-68	-28
BzHis	227	114	128	0	60	100
f3Leu	167	54	68	-60	0	40
tbu-ala	127	14	28	-100	-40	0

Regarding Figure 5d, the triplicate experiments that were done for this experiment had masses of 1624.52, 1624.77 and 1624.93, as compared to the calculated mass of 1625.77. (The other MALDIs are in the SI, Figure 12). The spectrum we chose to show in the figure happened to have the most divergence from the expected mass. We've listed below the possible misincorporations for the CUU, CUG, and CUA codons, which were expected to encode for tBu-

Ala, Val, and norval, even including potential misincorporations with the canonical AAs used in creation of the tag. None of these potential misincorporations lead to masses of -1 Da relative to the expected mass.

		tbu-ala	Val	norval
	mass	127	99	99
tbu-ala	127	0	28	28
Val	99	-28	0	0
norval	99	-28	0	0
Lys	128	1	29	29
Met	131	4	32	32
Tyr	163	36	64	64
Asp	115	-12	16	16

Taken together, while there is some deviation from the expected mass in these experiments due to the external calibration of our MALDI, the masses are still consistent with the correct products being formed every case.

- T7 tRNA was chosen over wt tRNA for reprogramming degenerate codons because of its observed better orthogonality. One concern with the T7 tRNA is the resultant decreased yield of the in vitro translated peptides, which may still contain byproduct peaks with quantities that are close to the detection limit and affect the orthogonality assessment. Could you comment on the peptide yield (e.g. Fig 3 and Fig 5) and the detection limit of your MALDI method?

T7 tRNAs are inferior to wt-tRNAs in translation. In fact, our recent paper, *NAR*, **2022**, *50*, 11734, used an isotopic competition assay that showed that T7 tRNA^{Leu} species can read a given codon around 20% as efficiently as wt-tRNAs when placed in competition with each other. This is typically compensated for by adding reasonably high concentrations (25 μ M or higher) of these AA-tRNAs in translation, particularly when they bear difficult ncAAs.

We have not measured peptide yields for Fig. 3, as our goal for this figure was to detect misincorporation events which cannot be detected using our typical ³⁵S-Met capture assay. We did measure yields for Fig. 5 (Fig. 5c), which incorporate all of the tested AA-tRNAs from the previous figures. These were reported relative to the wild-type yields. We have now added a new SI figure 11 showing the absolute yields for these experiments.

With our MALDI assay, we can typically detect peptides that are present at quantities about 50 fmols. Our typical yield for these in vitro translation assays are between 1-10 pmols, which means we can detect peaks at the 0.5-5% range.

- This is an intensively studied field filled with controversial observations. In Suga's paper (*Nat Chem.* 2016 Apr;8(4):317-25; Fig 3), his group showed that tRNA-Val-CAC can specifically target GUG codon but not GUC. However, the current manuscript showed a significant crosstalk between tRNA-Val-CAC and the GUC codon (Omer-Thr readthrough, Fig S9). Could you address these inconsistent findings?

This is an astute observation. The mutant tRNA^{Val}_{CAC} in our hands was able to read the GUC codon, but in Suga's work it was orthogonal to tRNA_{GAC} which would be expected to read the

GUC codon. There are some differences between our study and Suga's that could be illustrative. First, Suga's study utilized the tRNA asparagine body (tRNA^{Asn}_{GAC}) instead of tRNA^{Val}_{GAC}. A second important difference is that the experiments described in Suga's paper utilize a different amino acid (N-methyl tyrosine on the tRNA^{AsnE2}_{GAC}) and they also add this AA-tRNA at an extremely high concentration (100 μM) while the Val tRNA^{Val}_{CAC} they used was present at 5 μM. The large difference in these concentrations suggests that Suga's experiments were highly optimized in order to get the orthogonal result. The tRNA studies in our paper, other than those involving BzHis, were all conducted with the tRNAs on equal footing and are likely to represent true orthogonality comparisons.

- To assess the freedom and establish the generality of assigning different non-natural amino acids to the chosen split codons, could you compare different permutations of the AA- tRNA pairing in Fig 3C and Fig 5B? For example, BzHis is the most "challenging" amino acid among all tested. Could it be only mapped to the CUG codon cleanly or does it get incorporated to other degenerate codons for Leucine equally well?

This is an interesting question. We did not carefully manipulate the AAs and tRNAs to get the results in the paper. The data described in Figure 3C and 5B resulted from a random choice about which tRNA to pair with which AA, and resulted from the first pairings we tried. To test other permutations, we have performed an additional experiment where we swapped the tRNAs containing BzHis and 3F-Leu to Leu_{UUA} and Leu_{CAG} respectively. This data is now included in Supplementary Figure 8. While the orthogonality is not as strong as the original pairings, the major peak is the expected mass (Figure S8A), with substitutions for BzHis being a key issue. As before, the misincorporations observed under the initial conditions also served to help us improve orthogonality by adjusting tRNA concentrations (Figure S8B). That some optimization is required is not surprising as these types of adjustments are often practiced even for tRNAs reading different codon boxes. In this particular case, we think that EF-Tu affinity may be a factor. The Leu_{UUA} tRNA which carried BzHis in the experiments described in Fig. S8 is predicted to have weaker affinity for EF-Tu based on its T-stem sequence as it contains a less favorable base pair (CG) in the 51:63 T-stem sequence as compared to the Leu_{CAG} tRNA (GC) which was used in the main paper figures (see Schrader et. al., *J. Mol. Biol.* (2009) 386, 1255-1264, for analysis of T-stem base pairs and EF-Tu affinity). The focus of our paper is not to show that orthogonality occurs automatically without needed optimization, but rather that by combining hyperaccurate ribosomes and T7 tRNAs, codon boxes can be broken to a greater extent than previously thought possible. Downstream studies, outside the scope of this paper, will afford new abilities to predict and optimize tRNA/AA pairings to harness this newfound orthogonality.

- Consider adding a paragraph to discuss the scope and/or the limitation of the non-natural amino acids that can be reprogrammed using this synthetase pre-charged method, in comparison to the wider range of monomers that can be utilized via flexizyme charging.

This is a valid point. We've added the following sentences to the discussion.

"This type of expansion requires the ability to charge these tRNAs with ncAAs. Here we focused on using the AARS enzymes for this purpose. A reasonably diverse set of ncAAs can be attached onto tRNAs using this approach; however, most of these are near-neighbors to the canonical AAs. A far greater diversity of ncAAs can be attached onto tRNAs in a tRNA agnostic

manner using the Flexizyme enzyme system. It should therefore be quite feasible to attach each of these tRNAs to different ncAAs, leading to a dramatically expanded code with exciting applications in the creation of diverse libraries using mRNA display.”

REVIEWERS' COMMENTS

Reviewer #1 (Remarks to the Author):

The authors satisfactorily addressed the minor concerns that I raised for the previous version of manuscript. Now, I recommend this manuscript for publication in Nature Communications.

Reviewer #2 (Remarks to the Author):

All questions and concerns raised in initial review have been addressed. I recommend publication.

Reviewer #3 (Remarks to the Author):

I was enthusiastic about this manuscript the first time and remain enthusiastic. The manuscript should be published and no additional significant modifications are needed.

Reviewer #4 (Remarks to the Author):

The authors have incorporated feedback from the reviewers and revised the manuscript accordingly, so I will recommend a publication.